# Syntheses of New Multisubstituted 1-Acyloxyindole Compounds

**DOI:** 10.3390/molecules27196769

**Published:** 2022-10-10

**Authors:** Ye Eun Kim, Yoo Jin Lim, Chorong Kim, Yu Ra Jeong, Hyunsung Cho, Sang Hyup Lee

**Affiliations:** College of Pharmacy and Innovative Drug Center, Duksung Women’s University, 33 Samyangro 144-gil, Dobong-gu, Seoul 01369, Korea

**Keywords:** 1-hydroxyindole, 1-acyloxyindole, tin(Ⅱ) chloride, conjugate nitrone, 1,8-diazabicyclo [5.4.0]undec-7-ene (DBU)

## Abstract

The syntheses of novel 1-acyloxyindole compounds **1** and the investigations on reaction pathways are presented. Nitro ketoester substrate **2**, obtained in a two-step synthetic process, underwent reduction, intramolecular addition, nucleophilic 1,5-addition, and acylation to afford 1-acyloxyindoles **1** in one pot. Based on the systematic studies, we established the optimized reaction conditions for **1** focusing on the final acylation step of the intermediate 1-hydroxyindole **8**. With the optimized conditions, we succeeded in synthesizing 21 examples of new 1-acyloxyindole derivatives **1** in modest yields (Y = 24 − 35%). Among the 1-acyloxyindole compounds, 1-acetoxyindole compounds **1x** were generally unstable, and their yields were relatively lower than the other 1-acyloxyindoles. We expect that a bulkier alkyl or aromatic group on R^2^ could stabilize the 1-acyloxyindole compounds. Significantly, one-pot reactions of a four-step sequence successfully generated compounds **1** that are all new and might be difficult to be synthesized otherwise.

## 1. Introduction

The indole structure is an important component of many biologically active natural and synthetic compounds. Indeed, 1-acyloxyindole compounds are one of the N(1)-substituted indole derivatives containing an acyloxy group (-OCOR) instead of hydrogen on N(1) position (Figure 1). Among 1-acyloxyindole compounds, 1-benzoyloxyindole was first reported by Acheson et al. in 1970 [1], and 1-acetoxyindole was then reported in 1974 [2]. Further, 1-hydroxyindole, 1-alkoxyindole, and 1-acyloxyindole derivatives have attracted attention by the emergence of several potent 1-hydroxyindole derivatives [3,4,5] and 1-methoxyindole derivatives [6,7] in natural products. Furthermore, these unique structures demonstrated potential as indole substitutes and indole precursors associated with metabolic transformations [8,9,10].

Several investigations reported that 1-hydroxyindole and 1-alkoxyindole compounds have various biological properties, including antitumor [11,12,13], antibiotic [14], and platelet aggregation inhibiting effects [15]. In particular, Granchi et al. discovered that 1-hydroxyindole derivatives inhibited lactate dehydrogenase isoform A (LDH-A) at low micromolar concentration [12] and thus were considered as potential compounds for cancer treatment [16]. Furthermore, these derivatives were employed as intermediates for synthesis of potent anticonvulsant and antiarrhythmic agents [17]. Despite their various biological activities, few studies on 1-hydroxy, 1-alkoxy, and 1-acyloxyindole derivatives have been conducted. Probably due to the instability of 1-hydroxyindole compounds [10,18], studies on synthetic methods and their derivatization have not been extensively explored. Somei et al. developed a synthetic method for structurally concise 1-hydroxyindoles by oxidation of 2,3-dihydroindoles with a catalytic amount of sodium tungstate (Na_2_WO_4_) [19,20]. Other groups employed Zn/NH_4_Cl [2], tin chloride (SnCl_2_) [21,22,23], or Pd/triethylammonium formate (TEAF) [24] for reductive cyclization to produce 1-hydroxyindole moiety, and 1-Alkoxyindole compounds were also synthesized by alkylation of 1-hydroxyindoles [25] or intramolecular cyclization of methoxime structure [26]. Although a few simple 1-acyloxyindole derivatives were synthesized by acylation of 1-hydroxyindoles [17,27,28,29,30], the generated 1-acyloxyindoles were unstable and easily hydrolyzed to afford 1-hydroxyindoles [31]. Consequently, these compounds suffer from chemical instabilities and difficult manipulation.

In this study, we aimed to create novel derivatives of multisubstituted 1-acyloxyindoles (Figure 2) with improved chemical stability and meaningful biological activity. With the nitro ketoester substrate obtained in a two-step sequence, we devised a convenient one-pot synthetic method of consecutive four-step sequence to afford the desired 1-acyloxyindole compounds **1**. In addition, we expect that the compounds could serve as useful prodrugs for valuable medicinal agents or pharmacokinetic structural components for drug delivery.

## 2. Results and Discussion

### 2.1. Synthesis of Conjugate Nitro Ketoester 2

At first, we prepared the substrate **2** [25,32,33,34] in two-step reactions using 2-chloro-6-nitrotoluene as a starting material. For this purpose, we applied our previous procedures [32] with minor modifications. As shown in Figure 1, 2-chloro-6-nitrotoluene **3** was reacted with dimethyloxalate in the presence of excess sodium hydride to afford **4** [32] in an excellent yield (Y = 96%). Subsequently, **4** was reacted with dimethylmethyleneimminium chloride to add a methylene group at α-carbon in **4**, affording conjugate ketoester **2** [32] in a good yield (Y = 85%). The result of synthesis of **2** was slightly improved compared to the previous results [32].

### 2.2. Optimization for Formation of 1-Acyloxyindoles 1

The reactions to generate 1-acyloxyindoles **1** consist of two main parts: formation of 1-hydroxyindole intermediates **8** and formation of 1-acyloxyindoles **1** by acylation of **8**. Although we previously established the reaction conditions for 1-hydroxyindoles **8** [32], re-optimization for synthesis of 1-acyloxyindoles **1** is required because the whole process, including the acylation step, needs to be carried out in one pot. With substrate **2**, we first attempted to perform systematic studies on the reaction conditions suitable for formation of 1-acyloxyindole **1**. As indicated in Figure 2, the substrate **2** was reduced to generate hydroxylamine **5**, cyclized to provide hydroxyindoline **6,** and dehydrated to produce conjugate nitrone **7**. Then, nucleophilic 1,5-addition of alcohol to **7** produced the intermediate 1-hydroxyindole **8**, and, finally, acylation of **8** afforded 1-acyloxyindole **1**.

In particular, as a base for the acylation reaction, we tested several reagents, such as K_2_CO_3_, triethylamine (TEA), *N*,*N*-diisopropylethylamine (DIEA), 4-dimethylaminopyridine (DMAP), and 1,8-diazabicyclo [5.4.0]undece-7-ene (DBU). Among them, DBU provided the best results (data not shown), which are consistent with our previous reports [25]. Thus, we chose DBU for our purpose. Optimization of the reaction conditions was performed by varying the amount of SnCl_2_∙2H_2_O, DBU, alcohol (R^1^OH), and acylating agent (R^2^COX). SnCl_2_∙2H_2_O, an appropriate reducing agent for aromatic nitro group [35], was applied to convert **2** to **7** (**2** → **5** → **6** → **7**). We used benzyl alcohol (BnOH) as a template nucleophile and pivaloyl chloride as a template acylating agent in dimethoxyethane (DME) to produce **1dy** (Table 1). Considering our previous procedure for synthesis of 1-alkoxyindoles [25], we applied the range of reagents as such: SnCl_2_∙2H_2_O 2.5–3.7 eq and DBU 10.6–15.7 eq for synthesis of **1**. Here, we used an increased amount of DBU due to expected extra consumption by carboxylic acids that could be generated by partial hydrolysis of the acylating agents. At lower or higher amounts than 3.3 eq for SnCl_2_∙2H_2_O and 14.0 eq for DBU, the product **1dy** was obtained in relatively poor yields (entries 1, 2, and 7, Table 1). We also compared the yields of 1-hydroxyindole intermediate **8d** by varying the amount of SnCl_2_∙2H_2_O and found that the isolated yield of 1-hydroxyindole **8d** with 3.3 eq of SnCl_2_∙2H_2_O was better (Y = 48%) than 3.7 eq (Y = 33%) and 2.5 eq (Y = 32%) in the case of 2.0 eq of BnOH. The orders of yields for intermediate 1-hydroxyindole **8d** and 1-pivaloyloxyindole **1dy** were generally correlated. The amount of SnCl_2_∙2H_2_O might be an important factor for construction of 1-acyloxyindole as well as 1-hydroxyindole intermediate by triggering the reduction of nitro group in **2**. We further tested the amount of BnOH (1.5–3.0 eq) and pivaloyl chloride (1.5–3.0 eq). When using BnOH less than 1.5 eq, the product **1dy** was obtained in poor yields (entries 3 and 4). More than 2.0 eq of BnOH and pivaloyl chloride did not seem to improve the yield (entry 6). Taken together, we chose the optimized condition for **1dy** (entry 5): 1.0 eq of **2**, 3.3 eq of SnCl_2_∙2H_2_O, 2.0 eq of BnOH at 40 °C, and then 14.0 eq of DBU and 2.0 eq of pivaloyl chloride at room temperature, which was applied to all other reactions unless otherwise noted.

### 2.3. Synthesis of New Derivatives of 1-Acyloxyindole 1

Under the optimized condition (entry 5, Table 1), we synthesized new 1-acyloxyindole derivatives by employing various nucleophiles and several acylating reagents (acetic anhydride and acyl chlorides) (Figure 3). First, SnCl_2_·2H_2_O and 4Å molecular sieves were stirred in DME for 30 min at room temperature. We added alcohol and substrate **2**, and then the reaction mixture was stirred at 40 °C for 1.5–3 h. After confirming that the starting material **2** was converted to 1-hydroxyindole **8** by checking TLC, we slowly added 14.0 eq of DBU with vigorous stirring. The reaction mixture was stirred for 30 min at room temperature and then acetic anhydride or acyl chloride was added in an ice bath. We kept stirring the reaction mixture at room temperature for 1.5–4 h, leading to formation of targeted 1-acyloxyindoles **1**.

As acylating agents of 1-hydroxyindole intermediate **8**, acetic anhydride, pivaloyl chloride, benzoyl chloride, butanoyl chloride, hexanoyl chloride, and hydrocinnamoyl chloride were employed (Table 2). For acetylation reactions, we used acetic anhydride instead of acetyl chloride due to the high reactivity and instability of acetyl chloride. For example, both acetic anhydride and acetyl chloride provided **1dx** in similar yields (Y = ~30%), so we chose acetic anhydride. The yields for acetylation were generally lower than those for pivaloylation and benzoylation. For example, among **1dx**, **1dy**, and **1dz** (entries 13–15), the yield of 1-acetoxyindole **1dx** was lower than those for **1dy** and **1dz** with bulkier alkyl and aromatic group, respectively. Moreover, the yield of **1dw** with phenethyl group (entry 12) was higher than that of **1dx**. We expected that low yields of 1-acetoxyindoles might be due to the instability of the compounds and that a bulkier alkyl or aromatic group on R^2^ could stabilize the 1-acyloxyindole compounds. Interestingly, when we analyzed the spectroscopic features of these compounds, we found some consistence. For example, we found that the δ values (^13^C NMR) of carbonyl carbons of N-O*C*(O)CH_3_ in 1-acetoxyindoles **1x** were ~168.5, which means an upfield shift (~2) compared with those of carbonyl carbons in corresponding esters (R-O*C*(O)CH_3_). In addition, the λ_max_ values in UV–Vis were in the range of 229–236 nm. We also performed some of the reactions for **1du**, **1dx**, and **1dz** in a larger scale (1.1 mmol of **2**) and confirmed robust reproducibility of the established optimized conditions. Consequently, we successfully synthesized 21 new 1-acyloxyindole compounds **1** in modest yields (Y = 24–35%).

Furthermore, some degree of decomposition of 1-acyloxyindoles (**1du**, **1dv**, and **1dx**) with linear alkyl groups (R^2^ = *n* − Pr, *n* − Pen, and Me) on a TLC plate was observed. Partial degradation was observed for 1-acetoxyindole **1dx** within 30 min, and, for 1-butanoyloxyindole **1du** and 1-hexanoyloxyindole **1dv,** within 2 h. However, 1-pivaloyloxyindole **1dy** and 1-benzoyloxyindole **1dz** were not easily decomposed on TLC. Consequently, we found that these 1-acyloxyindole compounds seem to exhibit significantly different stabilities depending on the R^2^ in the acyl group (R^2^CO). Furthermore, these observations prompted us to test the stability of these compounds under hydrolysis conditions. We found that 1-butanoyloxyindole **1du** and 1-acetoxyindole **1dx** were easily hydrolyzed to provide 1-hydroxyindole under mildly basic conditions (data not shown). It is expected that this instability is due to the labile ester bond of NO-C(O)R^2^. This bond seems easily cleavable in even weakly acidic or basic conditions, resulting in 1-hydroxyindole and carboxylic acids (Figure 4). We believed that this labile ester bond might provide us with an interesting possibility of its application in a prodrug strategy, which aims to explore drug delivery by lowering the polarity of the compounds by acylation of 1-hydroxyindole. Thus, further application studies on stability are in progress.

### 2.4. Mechanistic Investigations on Reaction Pathways

We investigated the reaction mechanisms and pathways based on the observed products, as shown in Figure 5. We suggest that three pathways, A_1_, A_2_, and B, are involved in the reaction mechanism, which derives some support from our previous work [34]. The nitro group of conjugate ketoester derivative **2** was reduced to afford hydroxylamine compound **5** (or conformer **5′**). The pathways A_1_ and A_2_ proceeded through conformer **5**, and pathway B through conformer **5′**. The intramolecular addition of N-H of two conformers, **5** and **5′**, provided two different indoline derivatives, **6** and **10,** respectively. Following dehydration of **6,** it was possible to generate conjugate nitrone **7**. Nucleophilic 1,5-addition of alcohol (R^1^OH) produced 1-hydroxyindole **8** (Path A_1_); subsequent acylation of the hydroxy group with R^2^COX provided 1-acyloxyindole **1**. However, instead of alcohol, H_2_O as a nucleophile could be added to conjugate nitrone **7** to produce dihydroxy species **9** (Path A_2_). Dihydroxy compound **9** could be acylated with R^2^COX to provide diacylated compound **12**. In the process of synthesizing **1dy**, dipivaloylated compound **12** (R^2^ = *t*-Bu) was obtained and identified by mass analysis (446 [M + Na]^+^). On the other hand, conformer **5′** produced enolic compound **10** through intramolecular conjugate addition (aza-Michael addition) (Path B). Then, subsequent oxidative aromatization provided 1-hydroxyindole **11** [34]. Although we expected that acylation of **11** could produce **13**, the acylated product **13** was difficult to be isolated and even identified. In most of the reactions in Table 2, we believed that substrate **2** proceeded through not only Path A_1_ but also Path A_2_ and Path B, which might explain the low yields of the products **1**.

## 3. Experimental

### 3.1. General

Reagents were obtained from Sigma-Aldrich (Darmstad, Germany), Thermo Fisher (Waltham, MA, USA), and TCI (Tokyo, Japan). They were of commercial quality and used without further purification unless otherwise stated. Reactions were periodically monitored by thin-layer chromatography (TLC) carried out on 0.25 mm Merck silica gel plates (20 × 20 cm; Merck F_254_) (Darmstad, Germany) and visualized by UV light. Purifications were performed by preparative TLC (PTLC) and column chromatography. PTLC separations were carried out on the same silica gel plates. Column chromatography was performed using Merck silica gels (230–400 mesh) (Zvornik, Bosnia and Herzegovina). Melting points (uncorrected) were determined in Deckgläser microscope cover glasses (Lauda-Königshofen, Germany) using a Thermo Scientific 00590Q apparatus (Dubuque, Iowa, USA). ^1^H (300 MHz) and ^13^C (75 MHz) NMR spectra were obtained by a Bruker DRX 300 spectrometer (Zürich, Switzerland), and chemical shifts (δ) are expressed with respect to tetramethylsilane (TMS). NMR spectra are presented in the Appendix A. Mass spectra were obtained in EI or ESI ionization modes (Agilent, Santa Clara, CA, USA). High resolution mass spectra were obtained using JEOL apparatus (Tokyo, Japan) at the Korea Basic Science Institute, Republic of Korea. HPLC analyses were performed using the following Waters Associate Units: 515 A pump, 515 B pump, dual λ absorbance 2487 detector, and COSMOSIL 5C_18_-AR-Ⅱ Packed Column (4.6 × 250 mm) (Worcester, MA, USA). The products were analyzed using a linear gradient: from 70% A (aqueous) and 30% B (acetonitrile) for 3 min (isocratic) to 10% A and 90% B over 30 min at a flow rate of 1 mL/min with eluent monitoring at 254 nm. HPLC solvents were filtered (aqueous solution with PALL FP-450, 0.45 μm, 47 mm; acetonitrile with PALL TF-450, 0.45 μm, 47 mm) and degassed before use.

### 3.2. Substrate Synthesis

Methyl 3-(2′-Chloro-6′-Nitrophenyl)-2-Oxopropanoate (**4**) [32]

2-Chloro-6-nitrotoluene (**3**, 1.17 g, 6.8 mmol, 1.0 eq) and dimethyl oxalate (4.02 g, 34.0 mmol, 5.0 eq) were dissolved in anhydrous DMF (8.2 mL). To a stirred mixture of NaH (60% in mineral oil, 1.09 g, 27.2 mmol, 4.0 eq) in anhydrous DMF (4.1 mL) at 0 °C was added dropwise a solution of dimethyl oxalate and 2-chloro-6-nitrotoluene. The reaction mixture was stirred at 0 °C for 1 h and at room temperature for 4.5 h. The reaction mixture was quenched with saturated NH_4_Cl (15 mL) at 0 °C, extracted with methylene chloride (2 × 50 mL), and washed with H_2_O (2 × 50 mL). The organic layer was dried over MgSO_4_ and concentrated. The residue was purified by column chromatography (1:4 → 1:2 EtOAc/hexanes) to obtain compound **4** (1.68 g, 96%) as a pale-yellow solid. Spectral data are in accordance with literature information [32].

Methyl 3-(2′-chloro-6′-nitrophenyl)-2-oxobut-3-enoate (**2**) [32]

Ketoester (**4**, 1.53 g, 5.95 mmol, 1.0 eq) was dissolved in anhydrous THF (50 mL). To a stirred mixture of NaH (60% in mineral oil, 262 mg, 6.54 mmol, 1.1 eq) in anhydrous THF (100 mL) at 0 °C was added dropwise a solution of ketoester. After stirring for 1 h at 0 °C, *N,N*-dimethylmethyleneiminium chloride (1.85 g, 17.84 mmol, 3.0 eq) was added and the reaction mixture was stirred for 1 h at 0 °C. The reaction mixture was allowed to warm to room temperature and stirred for additional 5 h. The reaction mixture was quenched with saturated NH_4_Cl (10 mL) at 0 °C, extracted with EtOAc (2 × 250 mL), and washed with H_2_O (2 × 250 mL). The organic layer was dried over MgSO_4_ and concentrated. The residue was purified by column chromatography (1:4 → 1:2 EtOAc/hexanes) to obtain compound **2** (1.36 g, 85%) as a pale-yellow solid. Spectral data are in accordance with literature information [32].

### 3.3. General Procedure for Synthesis of 1-Acyloxyindoles 1

SnCl_2_·2H_2_O and 4Å molecular sieves stirred in DME for 30 min at room temperature. To a stirred mixture was added alcohol and conjugate ketoester **2**. The resulting mixture was stirred for 1.5–3 h at 40 °C. After confirming that the starting material was disappeared by using TLC, DME and DBU were added and stirred strongly for 30 min at room temperature. The acetic anhydride or acyl chloride was added in ice bath and kept stirring at room temperature for 1.5–4 h until reaction ends. The reaction mixture was diluted with CH_2_Cl_2_ and washed with diluted water, saturated aqueous ammonium chloride, sodium bicarbonate, and brine. The organic layer was dried over MgSO_4_ and concentrated. The residue was purified by preparative TLC (PTLC) and column chromatography to provide 1-acyloxyindoles **1**.

Methyl 4-chloro-1-acetoxy-3-[(methoxy)methyl]-1H-indole-2-carboxylate (**1ax**)

Use of SnCl_2_·2H_2_O (82.8 mg, 0.37 mmol, 3.3 eq), methanol (9 μL, 0.22 mmol, 2.0 eq), and **2** (30 mg, 0.11 mmol, 1.0 eq) for 2.5 h at 40 °C, then use of DBU (233 μL, 1.56 mmol, 14.0 eq) and acetic anhydride (32 μL, 0.22 mmol, 2.0 eq) for 4 h in general procedure afforded the title compound **1ax** (9.6 mg, 28%) as a yellow solid. Mp 104–106 °C; *R*_f_ 0.35 (1:2 EtOAc/hexanes); HPLC t*_R_* 12.1 min; UV–Vis (CH_3_CN-H_2_O) λ_max_ 213, 232, 296 nm; ^1^H NMR (300 MHz, CDCl_3_) *δ* 7.30–7.14 (m, 3H, Ar), 5.12 (s, 2H, C(3)CH_2_O), 3.95 (s, 3H, CO_2_CH_3_), 3.46 (s, 3H, CH_2_OC*H*_3_), 2.43 (s, 3H, OC(O)CH_3_); ^13^C NMR (75 MHz, CDCl_3_) *δ* 168.5 (NOC(O)), 160.4 (*C*O_2_CH_3_), 136.9, 128.9, 127.1, 125.0, 123.7, 120.1, 118.2, 108.0 (Ar), 63.5 (CH_2_O*C*H_3_), 58.1 (C(3)*C*H_2_O), 52.4 (CO_2_*C*H_3_), 18.2 (OC(O)*C*H_3_); MS m/z 311 [M]^+^; HRMS (+ESI) calcd for C_14_H_14_ClNO_5_ [M]^+^ 311.0561, found 311.0559.

Methyl 4-chloro-1-pivaloyloxy-3-[(methoxy)methyl]-1H-indole-2-carboxylate (**1ay**)

Use of SnCl_2_·2H_2_O (82.8 mg, 0.37 mmol, 3.3 eq), methanol (9 μL, 0.22 mmol, 2.0 eq), and **2** (30 mg, 0.11 mmol, 1.0 eq) for 1.5 h at 40 °C, then use of DBU (233 μL, 1.56 mmol, 14.0 eq) and pivaloyl chloride (28 μL, 0.22 mmol, 2.0 eq) for 2 h in general procedure afforded the title compound **1ay** (14.1 mg, 35%) as a pale-yellow solid. Mp 88–89 °C; *R*_f_ 0.47 (1:2 EtOAc/hexanes); HPLC t*_R_* 37.9 min; UV–Vis (CH_3_CN-H_2_O) λ_max_ 235, 296 nm; ^1^H NMR (300 MHz, CDCl_3_) *δ* 7.28–7.18 (m, 2H, Ar), 7.06 (dd, *J* = 7.8, 0.9 Hz, 1H, Ar), 5.13 (s, 2H, C(3)CH_2_O), 3.92 (s, 3H, CO_2_CH_3_), 3.44 (s, 3H, CH_2_OC*H*_3_), 1.47 (s, 9H, C(CH_3_)_3_); ^13^C NMR (75 MHz, CDCl_3_) *δ* 175.7 (NOC(O)), 160.2 (*C*O_2_CH_3_), 137.0, 128.9, 127.0, 125.7, 123.6, 120.3, 118.3, 107.8 (Ar), 63.5 (CH_2_O*C*H_3_), 58.0 (C(3)*C*H_2_O), 52.3 (CO_2_*C*H_3_), 38.8 (*C*(CH_3_)_3_), 27.4 (C(*C*H_3_)_3_); MS m/z 353 [M]^+^; HRMS (+ESI) calcd for C_17_H_20_ClNO_5_ [M]^+^ 353.1030, found 353.1028.

Methyl 4-chloro-1-benzoyloxy-3-[(methoxy)methyl]-1H-indole-2-carboxylate (**1az**)

Use of SnCl_2_·2H_2_O (82.8 mg, 0.37 mmol, 3.3 eq), methanol (9 μL, 0.22 mmol, 2.0 eq), and **2** (30 mg, 0.11 mmol, 1.0 eq) for 1.5 h at 40 °C, then use of DBU (233 μL, 1.56 mmol, 14.0 eq) and benzoyl chloride (26 μL, 0.22 mmol, 2.0 eq) for 2 h in general procedure afforded the title compound **1az** (13.5 mg, 33%) as a pale-yellow solid. Mp 112–114 °C; *R*_f_ 0.39 (1:2 EtOAc/hexanes); HPLC t*_R_* 37.1 min; UV–Vis (CH_3_CN-H_2_O) λ_max_ 236, 296 nm; ^1^H NMR (300 MHz, CDCl_3_) *δ* 8.22 (d, *J* = 7.6 Hz, 2H, Ar), 7.72 (t, *J* = 7.4 Hz, 1H, Ar), 7.57 (t, *J* = 7.7 Hz, 2H, Ar), 7.29–7.19 (m, 3H, Ar), 5.18 (s, 2H, C(3)CH_2_O), 3.83 (s, 3H, CO_2_CH_3_), 3.48 (s, 3H, CH_2_OC*H*_3_); ^13^C NMR (75 MHz, CDCl_3_) *δ* 164.6 (NOC(O)), 160.3 (*C*O_2_CH_3_), 137.4, 134.9, 130.6, 129.2, 128.9, 127.2, 126.4, 125.5, 123.8, 120.4, 118.8, 108.3 (Ar), 63.5 (CH_2_O*C*H_3_), 58.1 (C(3)*C*H_2_O), 52.4 (CO_2_*C*H_3_); MS m/z 373 [M]^+^; HRMS (+ESI) calcd for C_19_H_16_ClNO_5_ [M]^+^ 373.0717, found 373.0718.

Methyl 4-chloro-1-acetoxy-3-[(*n*-butyloxy)methyl]-1H-indole-2-carboxylate (**1bx**)

Use of SnCl_2_·2H_2_O (82.8 mg, 0.37 mmol, 3.3 eq), *n*-butanol (21 μL, 0.22 mmol, 2.0 eq), and **2** (30 mg, 0.11 mmol, 1.0 eq) for 1.5 h at 40 °C, then use of DBU (233 μL, 1.56 mmol, 14.0 eq) and acetic anhydride (32 μL, 0.22 mmol, 2.0 eq) for 2 h in general procedure afforded the title compound **1bx** (9.7 mg, 25%) as a pale-yellow solid. Mp 48–50 °C; *R*_f_ 0.26 (1:4 EtOAc/hexanes); HPLC t*_R_* 27.9 min; UV–Vis (CH_3_CN-H_2_O) λ_max_ 234, 296 nm; ^1^H NMR (300 MHz, CDCl_3_) *δ* 7.32–7.13 (m, 3H, Ar), 5.11 (s, 2H, C(3)CH_2_O), 3.94 (s, 3H, CO_2_CH_3_), 3.59 (t, *J* = 6.4 Hz, 2H, OC*H*_2_CH_2_), 2.43 (s, 3H, OC(O)CH_3_), 1.60 (quintet, *J* = 7.0 Hz, 2H, OCH_2_C*H*_2_), 1.39 (sextet, *J* = 7.4 Hz, 2H, O(CH_2_)_2_C*H*_2_), 0.89 (t, *J* = 7.2 Hz, 3H, O(CH_2_)_3_C*H*_3_); ^13^C NMR (75 MHz, CDCl_3_) *δ* 168.5 (NOC(O)), 160.4 (*C*O_2_CH_3_), 137.1, 129.0, 127.0, 125.0, 123.6, 120.3, 118.6, 108.0 (Ar), 70.4 (O*C*H_2_CH_2_), 62.0 (C(3)*C*H_2_O), 52.4 (CO_2_*C*H_3_), 32.1(OCH_2_*C*H_2_), 19.6 (O(CH_2_)_2_*C*H_2_), 18.2 (OC(O)*C*H_3_) 14.1 (O(CH_2_)_3_*C*H_3_); MS m/z 353 [M]^+^; HRMS (+ESI) calcd for C_17_H_20_ClNO_5_ [M]^+^ 353.1030, found 353.1031.

Methyl 4-chloro-1-pivaloyloxy-3-[(*n*-butyloxy)methyl]-1H-indole-2-carboxylate (**1by**)

Use of SnCl_2_·2H_2_O (82.8 mg, 0.37 mmol, 3.3 eq), *n*-butanol (21 μL, 0.22 mmol, 2.0 eq), and **2** (30 mg, 0.11 mmol, 1.0 eq) for 1.5 h at 40 °C, then use of DBU (233 μL, 1.56 mmol, 14.0 eq) and pivaloyl chloride (28 μL, 0.22 mmol, 2.0 eq) for 2 h in general procedure afforded the title compound **1by** (12.0 mg, 28%) as a pale-yellow solid. Mp 66–68 °C; *R*_f_ 0.53 (1:4 EtOAc/hexanes); HPLC t*_R_* 33.3 min; UV–Vis (CH_3_CN-H_2_O) λ_max_ 211, 230, 296 nm; ^1^H NMR (300 MHz, CDCl_3_) *δ* 7.27–7.17 (m, 2H, Ar), 7.04 (d, *J* = 7.8 Hz, 1H, Ar), 5.13 (s, 2H, C(3)CH_2_O), 3.91 (s, 3H, CO_2_CH_3_), 3.58 (t, *J* = 6.5 Hz, 2H, OC*H*_2_CH_2_), 1.68–1.32 (m, 4H, OCH_2_(C*H*_2_)_2_), 1.47 (s, 9H, C(CH_3_)_3_), 0.89 (t, *J* = 7.3 Hz, 3H, O(CH_2_)_3_C*H*_3_); ^13^C NMR (75 MHz, CDCl_3_) *δ* 175.7 (NOC(O)), 160.3 (*C*O_2_CH_3_), 137.2, 129.0, 126.9, 125.7, 123.5, 120.4, 118.8, 107.8 (Ar), 70.3 (O*C*H_2_CH_2_), 61.9 (C(3)*C*H_2_O), 52.3 (CO_2_*C*H_3_), 38.8 (*C*(CH_3_)_3_), 32.1 (OCH_2_*C*H_2_), 27.4 (C(*C*H_3_)_3_), 19.6 (O(CH_2_)_2_*C*H_2_), 14.1 (O(CH_2_)_3_*C*H_3_); MS m/z 395 [M]^+^; HRMS (+ESI) calcd for C_20_H_26_ClNO_5_ [M]^+^ 395.1500, found 395.1500.

Methyl 4-chloro-1-benzoyloxy-3-[(*n*-butyloxy)methyl]-1H-indole-2-carboxylate (**1bz**)

Use of SnCl_2_·2H_2_O (82.8 mg, 0.37 mmol, 3.3 eq), *n*-butanol (21 μL, 0.22 mmol, 2.0 eq), and **2** (30 mg, 0.11 mmol, 1.0 eq) for 1.5 h at 40 °C, then use of DBU (233 μL, 1.56 mmol, 14.0 eq) and benzoyl chloride (26 μL, 0.22 mmol, 2.0 eq) for 2 h in general procedure afforded the title compound **1bz** (14.4 mg, 32%) as a pale-yellow solid. Mp 56–58 °C; *R*_f_ 0.38 (1:4 EtOAc/hexanes); HPLC t*_R_* 32.6 min; UV–Vis (CH_3_CN-H_2_O) λ_max_ 229, 296 nm; ^1^H NMR (300 MHz, CDCl_3_) *δ* 8.22 (d, *J* = 7.3 Hz, 2H, Ar), 7.72 (t, *J* = 7.5 Hz, 1H, Ar), 7.57 (t, *J* = 7.8 Hz, 2H, Ar), 7.29–7.18 (m, 3H, Ar), 5.17 (s, 2H, C(3)CH_2_O), 3.83 (s, 3H, CO_2_CH_3_), 3.62 (t, *J* = 6.5 Hz, 2H, OC*H*_2_CH_2_), 1.63 (quintet, *J* = 6.5 Hz, 2H, OCH_2_C*H*_2_), 1.41 (sextet, *J* = 7.2 Hz, 2H, O(CH_2_)_2_C*H*_2_), 0.91 (t, *J* = 7.3 Hz, 3H, O(CH_2_)_3_C*H*_3_); ^13^C NMR (75 MHz, CDCl_3_) *δ* 164.6 (NOC(O)), 160.3 (*C*O_2_CH_3_), 137.5, 134.9, 130.6, 129.2, 129.0, 127.1, 126.3, 125.5, 123.7, 120.5, 119.2, 108.2 (Ar), 70.4 (O*C*H_2_CH_2_), 61.9 (C(3)*C*H_2_O), 52.3 (CO_2_*C*H_3_), 32.1 (OCH_2_*C*H_2_), 19.6 (O(CH_2_)_2_*C*H_2_), 14.1 (O(CH_2_)_3_*C*H_3_); MS m/z 415 [M]^+^; HRMS (+ESI) calcd for C_22_H_22_ClNO_5_ [M]^+^ 415.1187, found 415.1185.

Methyl 4-chloro-1-acetoxy-3-[(*n*-hexyloxy)methyl]-1H-indole-2-carboxylate (**1cx**)

Use of SnCl_2_·2H_2_O (82.8 mg, 0.37 mmol, 3.3 eq), *n*-hexanol (44 μL, 0.22 mmol, 2.0 eq), and **2** (30 mg, 0.11 mmol, 1.0 eq) for 2 h at 40 °C, then use of DBU (233 μL, 1.56 mmol, 14.0 eq) and acetic anhydride (32 μL, 0.22 mmol, 2.0 eq) for 2 h in general procedure afforded the title compound **1cx** (11.3 mg, 27%) as a pale-yellow solid. Mp 51–53 °C; *R*_f_ 0.26 (1:4 EtOAc/hexanes); HPLC t*_R_* 31.4 min; UV–Vis (CH_3_CN-H_2_O) λ_max_ 235, 296 nm; ^1^H NMR (300 MHz, CDCl_3_) *δ* 7.29–7.19 (m, 2H, Ar), 7.14 (dd, *J* = 7.1, 0.9 Hz, 1H, Ar), 5.12 (s, 2H, C(3)CH_2_O), 3.94 (s, 3H, CO_2_CH_3_), 3.58 (t, *J* = 6.6 Hz, 2H, OC*H*_2_CH_2_), 2.42 (s, 3H, OC(O)CH_3_), 1.62 (quintet, *J* = 6.8 Hz, 2H, OCH_2_C*H*_2_), 1.40–1.25 (m, 6H, O(CH_2_)_2_(C*H*_2_)_3_), 0.86 (t, *J* = 6.5 Hz, 3H, O(CH_2_)_5_C*H*_3_); ^13^C NMR (75 MHz, CDCl_3_) *δ* 168.5 (NOC(O)), 160.4 (*C*O_2_CH_3_), 137.1, 129.0, 127.0, 125.0, 123.6, 120.3, 118.6, 108.0 (Ar), 70.7 (O*C*H_2_CH_2_), 62.0 (C(3)*C*H_2_O), 52.4 (CO_2_*C*H_3_), 31.9 (OCH_2_*C*H_2_), 30.0 (O(CH_2_)_2_*C*H_2_), 26.1 (O(CH_2_)_3_*C*H_2_), 22.8 (O(CH_2_)_4_*C*H_2_), 18.2 (OC(O)*C*H_3_), 14.3 (O(CH_2_)_5_*C*H_3_); MS m/z 381 [M]^+^; HRMS (+ESI) calcd for C_19_H_24_ClNO_5_ [M]^+^ 381.1343, found 381.1339.

Methyl 4-chloro-1-pivaloyloxy-3-[(*n*-hexyloxy)methyl]-1H-indole-2-carboxylate (**1cy**)

Use of SnCl_2_·2H_2_O (82.8 mg, 0.37 mmol, 3.3 eq), *n*-hexanol (44 μL, 0.22 mmol, 2.0 eq), and **2** (30 mg, 0.11 mmol, 1.0 eq) for 2 h at 40 °C, then use of DBU (233 μL, 1.56 mmol, 14.0 eq) and pivaloyl chloride (28 μL, 0.22 mmol, 2.0 eq) for 1.5 h in general procedure afforded the title compound **1cy** (13.7 mg, 29%) as a pale-yellow solid. Mp 57–58 °C; *R*_f_ 0.55 (1:4 EtOAc/hexanes); HPLC t*_R_* 36.5 min; UV–Vis (CH_3_CN-H_2_O) λ_max_ 235, 296 nm; ^1^H NMR (300 MHz, CDCl_3_) *δ* 7.27–7.18 (m, 2H, Ar), 7.05 (dd, *J* = 7.9, 0.9 Hz, 1H, Ar), 5.14 (s, 2H, C(3)CH_2_O), 3.92 (s, 3H, CO_2_CH_3_), 3.57 (t, *J* = 6.6 Hz, 2H, OC*H*_2_CH_2_), 1.68–1.25 (m, 8H, OCH_2_(C*H*_2_)_4_), 1.47 (s, 9H, C(CH_3_)_3_), 0.86 (t, *J* = 6.6 Hz, 3H, O(CH_2_)_5_C*H*_3_); ^13^C NMR (75 MHz, CDCl_3_) *δ* 175.6 (NOC(O)), 160.3 (*C*O_2_CH_3_), 137.2, 129.0, 126.9, 125.7, 123.5, 120.5, 118.8, 107.8 (Ar), 70.5 (O*C*H_2_CH_2_), 61.9 (C(3)*C*H_2_O), 52.2 (CO_2_*C*H_3_), 38.8 (*C*(CH_3_)_3_), 31.9 (OCH_2_C*H*_2_), 30.0 (O(CH_2_)_2_C*H*_2_), 27.4 (C(*C*H_3_)_3_), 26.1 (O(CH_2_)_3_*C*H_2_), 22.8 (O(CH_2_)_4_*C*H_2_), 14.2 (O(CH_2_)_5_*C*H_3_); MS m/z 423 [M]^+^; HRMS (+ESI) calcd for C_22_H_30_ClNO_5_ [M]^+^ 423.1813, found 423.1815.

Methyl 4-chloro-1-benzoyloxy-3-[(*n*-hexyloxy)methyl]-1H-indole-2-carboxylate (**1cz**)

Use of SnCl_2_·2H_2_O (82.8 mg, 0.37 mmol, 3.3 eq), *n*-hexanol (44 μL, 0.22 mmol, 2.0 eq), and **2** (30 mg, 0.11 mmol, 1.0 eq) for 2 h at 40 °C, then use of DBU (233 μL, 1.56 mmol, 14.0 eq) and benzoyl chloride (26 μL, 0.22 mmol, 2.0 eq) for 1 h in general procedure afforded the title compound **1cz** (14.2 mg, 30%) as a pale-yellow solid. Mp 62–64 °C; *R*_f_ 0.44 (1:4 EtOAc/hexanes); HPLC t*_R_* 35.4 min; UV–Vis (CH_3_CN-H_2_O) λ_max_ 235, 296 nm; ^1^H NMR (300 MHz, CDCl_3_) *δ* 8.21 (d, *J* = 7.4 Hz, 2H, Ar), 7.72 (t, *J* = 7.4 Hz, 1H, Ar), 7.57 (t, *J* = 7.9 Hz, 2H, Ar), 7.28–7.17 (m, 3H, Ar), 5.17 (s, 2H, C(3)CH_2_O), 3.82 (s, 3H, CO_2_CH_3_), 3.60 (t, *J* = 6.6 Hz, 2H, OC*H*_2_CH_2_), 1.63 (quintet, *J* = 7.0 Hz, 2H, OCH_2_C*H*_2_), 1.41–1.22 (m, 6H, O(CH_2_)_2_(C*H*_2_)_3_), 0.86 (t, *J* = 6.6 Hz, 3H, O(CH_2_)_5_C*H*_3_); ^13^C NMR (75 MHz, CDCl_3_) *δ* 164.7 (NOC(O)), 160.4 (*C*O_2_CH_3_), 137.6, 134.9, 130.6, 129.2, 129.0, 127.1, 126.4, 126.0, 123.7, 120.5, 119.3, 108.3 (Ar), 70.7 (O*C*H_2_CH_2_), 61.9 (C(3)*C*H_2_O), 52.3 (CO_2_*C*H_3_), 31.9 (OCH_2_*C*H_2_), 30.0 (O(CH_2_)_2_*C*H_2_), 26.1 (O(CH_2_)_3_*C*H_2_), 22.9 (O(CH_2_)_4_*C*H_2_), 14.3 (O(CH_2_)_5_*C*H_3_); MS m/z 443 [M]^+^; HRMS (+ESI) calcd for C_24_H_26_ClNO_5_ [M]^+^ 443.1500, found 443.1500.

Methyl 4-chloro-1-butanoyloxy-3-[(benzyloxy)methyl]-1H-indole-2-carboxylate (**1du**)

Use of SnCl_2_·2H_2_O (138 mg, 0.62 mmol, 3.3 eq), benzyl alcohol (38 μL, 0.37 mmol, 2.0 eq), and **2** (50 mg, 0.18 mmol, 1.0 eq) for 2.5 h at 40 °C, then use of DBU (388 μL, 2.60 mmol, 14.0 eq) and butanoyl chloride (38 μL, 0.37 mmol, 2.0 eq) for 2 h in general procedure afforded the title compound **1du** (20.4 mg, 26%) as a white solid. Mp 81–83 °C; *R*_f_ 0.56 (1:2 EtOAc/hexanes); HPLC t*_R_* 30.2 min; UV–Vis (CH_3_CN-H_2_O) λ_max_ 214, 234, 297 nm; ^1^H NMR (300 MHz, CDCl_3_) *δ* 7.42–7.20 (m, 7H, Ar), 7.12 (d, *J* = 7.9 Hz, 1H, Ar), 5.20 (s, 2H, C(3)CH_2_O), 4.67 (s, 2H, OC*H*_2_Ph), 3.82 (s, 3H, CO_2_CH_3_), 2.68 (t, *J* = 7.4 Hz, 2H, OC(O)CH_2_), 1.86 (sextet, *J* = 7.4 Hz, 2H, OC(O)CH_2_C*H*_2_), 1.10 (t, *J* = 7.4 Hz, 3H, OC(O)(CH_2_)_2_C*H*_3_); ^13^C NMR (75 MHz, CDCl_3_) *δ* 171.2 (NOC(O)), 160.3 (*C*O_2_CH_3_), 138.7, 137.0, 128.9, 128.5, 128.2, 127.7, 127.0, 125.2, 123.6, 120.2, 117.9, 108.0 (Ar), 72.6 (O*C*H_2_Ph), 61.7 (C(3)*C*H_2_O), 52.3 (CO_2_*C*H_3_), 33.4 (OC(O)*C*H_2_), 18.3 (OC(O)CH_2_*C*H_2_), 13.9 (OC(O)(CH_2_)_2_*C*H_3_); MS m/z 415 [M]^+^; HRMS (+ESI) calcd for C_22_H_22_ClNO_5_ [M]^+^ 415.1187, found 415.1184.

Methyl 4-chloro-1-hexanoyloxy-3-[(benzyloxy)methyl]-1H-indole-2-carboxylate (**1dv**)

Use of SnCl_2_·2H_2_O (138 mg, 0.61 mmol, 3.3 eq), benzyl alcohol (38 μL, 0.37 mmol, 2.0 eq), and **2** (50 mg, 0.18 mmol, 1.0 eq) for 2.5 h at 40 °C, then use of DBU (388 μL, 2.60 mmol, 14.0 eq) and hexanoyl chloride (52 μL, 0.37 mmol, 2.0 eq) for 2 h in general procedure afforded the title compound **1dv** (23.9 mg, 30%) as a pale-yellow solid. Mp 64–66 °C; *R*_f_ 0.62 (1:2 EtOAc/hexanes); HPLC t*_R_* 34.0 min; UV–Vis (CH_3_CN-H_2_O) λ_max_ 211, 234, 296 nm; ^1^H NMR (300 MHz, CDCl_3_) *δ* 7.41–7.20 (m, 7H, Ar), 7.11 (dd, *J* = 7.9, 1.2 Hz, 1H, Ar), 5.20 (s, 2H, C(3)CH_2_O), 4.66 (s, 2H, OC*H*_2_Ph), 3.82 (s, 3H, CO_2_CH_3_), 2.68 (t, *J* = 7.5 Hz, 2H, OC(O)CH_2_), 1.83 (quintet, *J* = 7.4 Hz, 2H, OC(O)CH_2_C*H*_2_), 1.49–1.34 (m, 4H, OC(O)(CH_2_)_2_(C*H*_2_)_2_), 0.94 (t, *J* = 6.9 Hz, 3H, OC(O)(CH_2_)_4_C*H*_3_); ^13^C NMR (75 MHz, CDCl_3_) *δ* 171.4 (NOC(O)), 160.3 (*C*O_2_CH_3_), 138.7, 137.1, 129.0, 128.5, 128.2, 127.7, 127.0, 125.9, 123.6, 120.3, 118.0, 108.0 (Ar) 72.6 (O*C*H_2_Ph), 61.7 (C(3)*C*H_2_O), 52.2 (CO_2_*C*H_3_), 31.5 (OC(O)*C*H_2_), 31.4 (OC(O)CH_2_*C*H_2_), 24.4 (OC(O)(CH_2_)_2_*C*H_2_), 22.4 (OC(O)(CH_2_)_3_*C*H_2_), 14.1 (OC(O)(CH_2_)_4_*C*H_3_); MS m/z 443 [M]^+^; HRMS (+ESI) calcd for C_24_H_26_ClNO_5_ [M]^+^ 443.1500, found 443.1496.

Methyl 4-chloro-1-hydrocinnamoyloxy-3-[(benzyloxy)methyl]-1H-indole-2-carboxylate (**1dw**)

Use of SnCl_2_·2H_2_O (138 mg, 0.61 mmol, 3.3 eq), benzyl alcohol (38 μL, 0.37 mmol, 2.0 eq), and **2** (50 mg, 0.18 mmol, 1.0 eq) for 3 h at 40 °C, then use of DBU (388 μL, 2.60 mmol, 14.0 eq) and hydrocinnamoyl chloride (55 μL, 0.37 mmol, 2.0 eq) for 3 h in general procedure afforded the title compound **1dw** (28.6 mg, 32%) as a pale-yellow solid. Mp 103–104 °C; *R*_f_ 0.48 (1:2 EtOAc/hexanes); HPLC t*_R_* 32.7 min; UV–Vis (CH_3_CN-H_2_O) λ_max_ 211, 235, 296 nm; ^1^H NMR (300 MHz, CDCl_3_) *δ* 7.41–7.14 (m, 13H, Ar), 5.19 (s, 2H, C(3)CH_2_O), 4.66 (s, 2H, OC*H*_2_Ph), 3.77 (s, 3H, CO_2_CH_3_), 3.14 (t, *J* = 7.1 Hz, 2H, (OC(O)CH_2_), 3.03 (t, *J* = 7.1 Hz, 2H, (OC(O)CH_2_C*H*_2_); ^13^C NMR (75 MHz, CDCl_3_) *δ* 170.6 (NOC(O)), 160.3 (*C*O_2_CH_3_), 139.7, 138.7, 137.0, 129.0, 128.8, 128.7, 128.5, 128.2, 127.7, 127.0, 126.9, 125.2, 123.6, 120.2, 118.1, 108.1 (Ar), 72.6 (O*C*H_2_Ph), 61.6 (C(3)*C*H_2_O), 52.2 (CO_2_*C*H_3_), 33.3 (OC(O)*C*H_2_), 30.7 (OC(O)CH_2_*C*H_2_); MS m/z 477 [M]^+^; HRMS (+ESI) calcd for C_27_H_24_ClNO_5_ [M]^+^ 477.1343, found 477.1345.

Methyl 4-chloro-1-acetoxy-3-[(benzyloxy)methyl]-1H-indole-2-carboxylate (**1dx**)

Use of SnCl_2_·2H_2_O (82.8 mg, 0.37 mmol, 3.3 eq), benzyl alcohol (24 μL, 0.22 mmol, 2.0 eq), and **2** (30 mg, 0.11 mmol, 1.0 eq) for 2 h at 40 °C, then use of DBU (233 μL, 1.56 mmol, 14.0 eq) and acetic anhydride (32 μL, 0.22 mmol, 2.0 eq) for 2 h in general procedure afforded the title compound **1dx** (13.0 mg, 30%) as a pale-yellow solid. Mp 68–69 °C; *R*_f_ 0.31 (1:2 EtOAc/hexanes); HPLC t*_R_* 26.6 min; UV–Vis (CH_3_CN-H_2_O) λ_max_ 236, 299 nm; ^1^H NMR (300 MHz, CDCl_3_) *δ* 7.42–7.14 (m, 8H, Ar), 5.20 (s, 2H, C(3)CH_2_O), 4.67 (s, 2H, OC*H*_2_Ph), 3.82 (s, 3H, CO_2_CH_3_), 2.42 (s, 3H, OC(O)CH_3_); ^13^C NMR (75 MHz, CDCl_3_) *δ* 168.5 (NOC(O)), 160.4 (*C*O_2_CH_3_), 138.6, 137.0, 129.0, 128.5, 128.2, 127.7, 127.1, 125.0, 123.7, 120.2, 118.0, 108.0 (Ar), 72.6 (O*C*H_2_Ph), 61.7 (C(3)*C*H_2_O), 52.3 (CO_2_*C*H_3_), 18.2 (OC(O)*C*H_3_); MS m/z 387 [M]^+^; HRMS (+ESI) calcd for C_20_H_18_ClNO_5_ [M]^+^ 387.0874, found 387.0875.

Methyl 4-chloro-1-pivaloyloxy-3-[(benzyloxy)methyl]-1H-indole-2-carboxylate (**1dy**)

Use of SnCl_2_·2H_2_O (82.8 mg, 0.37 mmol, 3.3 eq), benzyl alcohol (24 μL, 0.22 mmol, 2.0 eq), and **2** (30 mg, 0.11 mmol, 1.0 eq) for 2 h at 40 °C, then use of DBU (233 μL, 1.56 mmol, 14.0 eq) and pivaloyl chloride (28 μL, 0.22 mmol, 2.0 eq) for 2 h in general procedure afforded the title compound **1dy** (15.7 mg, 33%) as a pale-yellow solid. Mp 80–82 °C; *R*_f_ 0.65 (1:2 EtOAc/hexanes); HPLC t*_R_* 31.6 min; UV–Vis (CH_3_CN-H_2_O) λ_max_ 213, 234, 296 nm; ^1^H NMR (300 MHz, CDCl_3_) *δ* 7.41–7.19 (m, 7H, Ar), 7.06 (d, *J* = 7.7 Hz, 1H, Ar), 5.22 (s, 2H, C(3)CH_2_O), 4.66 (s, 2H, OC*H*_2_Ph), 3.82 (s, 3H, CO_2_CH_3_), 1.47 (s, 9H, C(CH_3_)_3_); ^13^C NMR (75 MHz, CDCl_3_) *δ* 175.7 (NOC(O)), 160.2 (*C*O_2_CH_3_), 138.8, 137.1, 129.0, 128.4, 128.2, 127.7, 126.9, 125.8, 123.6, 120.4, 118.2, 107.8 (Ar), 72.5 (O*C*H_2_Ph), 61.7 (C(3)*C*H_2_O), 52.2 (CO_2_*C*H_3_), 38.8 (O*C*(CH_3_)_3_), 27.4 (OC(*C*H_3_)_3_); MS m/z 429 [M]^+^; HRMS (+ESI) calcd for C_23_H_24_ClNO_5_ [M]^+^ 429.1343, found 429.1342.

Methyl 4-chloro-1-benzoyloxy-3-[(benzyloxy)methyl]-1H-indole-2-carboxylate (**1dz**)

Use of SnCl_2_·2H_2_O (82.8 mg, 0.37 mmol, 3.3 eq), benzyl alcohol (24 μL, 0.22 mmol, 2.0 eq), and **2** (30 mg, 0.11 mmol, 1.0 eq) for 2 h at 40 °C, then use of DBU (233 μL, 1.56 mmol, 14.0 eq) and benzoyl chloride (26 μL, 0.22 mmol, 2.0 eq) for 2 h in general procedure afforded the title compound **1dz** (16.5 mg, 33%) as a pale-yellow solid. Mp 112–114 °C; *R*_f_ 0.50 (1:2 EtOAc/hexanes); HPLC t*_R_* 31.0 min; UV–Vis (CH_3_CN-H_2_O) λ_max_ 235, 296 nm; ^1^H NMR (300 MHz, CDCl_3_) *δ* 8.20 (d, *J* = 7.2 Hz, 2H, Ar), 7.70 (t, *J* = 7.5 Hz, 1H, Ar), 7.55 (t, *J* = 7.9 Hz, 2H, Ar), 7.42–7.17 (m, 8H, Ar), 5.25 (s, 2H, C(3)CH_2_O), 4.68 (s, 2H, OC*H*_2_Ph), 3.72 (s, 3H, CO_2_CH_3_); ^13^C NMR (75 MHz, CDCl_3_) *δ* 164.6 (NOC(O)), 160.3 (*C*O_2_CH_3_), 138.7, 137.5, 134.9, 130.6, 129.2, 129.0, 128.2, 128.1, 127.7, 127.1, 126.7, 125.6, 123.7, 120.5, 118.6, 108.3 (Ar), 72.6 (O*C*H_2_Ph), 61.7 (C(3)*C*H_2_O), 52.3 (CO_2_*C*H_3_); MS m/z 449 [M]^+^; HRMS (+ESI) calcd for C_25_H_20_ClNO_5_ [M]^+^ 449.1030, found 449.1028.

Methyl 4-chloro-1-acetoxy-3-[(phenylethyloxy)methyl]-1H-indole-2-carboxylate (**1ex**)

Use of SnCl_2_·2H_2_O (82.8 mg, 0.37 mmol, 3.3 eq), 2-phenylethyl alcohol (27 μL, 0.22 mmol, 2.0 eq), and **2** (30 mg, 0.11 mmol, 1.0 eq) for 2 h at 40 °C, then use of DBU (233 μL, 1.56 mmol, 14.0 eq) and acetic anhydride (32 μL, 0.22 mmol, 2.0 eq) for 2 h in general procedure afforded the title compound **1ex** (11.5 mg, 26%) as a pale-yellow solid. Mp 70–71 °C; *R*_f_ 0.21 (1:4 EtOAc/hexanes); HPLC t*_R_* 27.6 min; UV–Vis (CH_3_CN-H_2_O) λ_max_ 211, 235, 296 nm; ^1^H NMR (300 MHz, CDCl_3_) *δ* 7.29–7.13 (m, 8H, Ar), 5.17 (s, 2H, C(3)CH_2_O), 3.89 (s, 3H, CO_2_CH_3_), 3.80 (t, *J* = 7.2 Hz, 2H, OC*H*_2_CH_2_), 2.94 (t, *J* = 7.4 Hz, 2H, OCH_2_C*H*_2_), 2.42 (s, 3H, OC(O)CH_3_); ^13^C NMR (75 MHz, CDCl_3_) *δ* 168.5 (NOC(O)), 160.4 (*C*O_2_CH_3_), 139.2, 137.0, 129.1, 129.0, 128.4, 127.0, 126.2, 125.0, 123.7, 120.2, 118.3, 108.0 (Ar), 71.4 (O*C*H_2_CH_2_), 62.1 (C(3)*C*H_2_O), 52.4 (CO_2_*C*H_3_), 36.5 (OCH_2_*C*H_2_), 18.2 (OC(O)*C*H_3_); MS m/z 401 [M]^+^; HRMS (+ESI) calcd for C_21_H_20_ClNO_5_ [M]^+^ 401.1030, found 401.1029.

Methyl 4-chloro-1-pivaloyloxy-3-[(phenylethyloxy)methyl]-1H-indole-2-carboxylate (**1ey**)

Use of SnCl_2_·2H_2_O (82.8 mg, 0.37 mmol, 3.3 eq), 2-phenylethyl alcohol (27 μL, 0.22 mmol, 2.0 eq), and **2** (30 mg, 0.11 mmol, 1.0 eq) for 2 h at 40 °C, then use of DBU (233 μL, 1.56 mmol, 14.0 eq) and pivaloyl chloride (28 μL, 0.22 mmol, 2.0 eq) for 1.5 h in general procedure afforded the title compound **1ey** (13.1 mg, 27%) as a pale-yellow solid. Mp 83–84 °C; *R*_f_ 0.59 (1:4 EtOAc/hexanes); HPLC t*_R_* 33.1 min; UV–Vis (CH_3_CN-H_2_O) λ_max_ 212, 235, 296 nm; ^1^H NMR (300 MHz, CDCl_3_) *δ* 7.32–7.18 (m, 7H, Ar), 7.05 (d, *J* = 8.0 Hz, 1H, Ar), 5.20 (s, 2H, C(3)CH_2_O), 3.88 (s, 3H, CO_2_CH_3_), 3.80 (t, *J* = 7.3 Hz, 2H, OC*H*_2_CH_2_), 2.94 (t, *J* = 7.4 Hz, 2H, OCH_2_C*H*_2_), 1.47 (s, 9H, C(CH_3_)_3_); ^13^C NMR (75 MHz, CDCl_3_) *δ* 175.7 (NOC(O)), 160.3 (*C*O_2_CH_3_), 139.3, 137.2, 129.2, 129.0, 128.5, 127.0, 126.4, 125.9, 123.6, 120.4, 118.6, 108.1 (Ar), 71.3 (O*C*H_2_CH_2_), 62.1 (C(3)*C*H_2_O), 52.3 (CO_2_*C*H_3_), 38.8 (O*C*(CH_3_)_3_), 36.6 (OCH_2_*C*H_2_), 27.4 (OC(*C*H_3_)_3_); MS m/z 443 [M]^+^; HRMS (+ESI) calcd for C_24_H_26_ClNO_5_ [M]^+^ 443.1500, found 443.1503.

Methyl 4-chloro-1-benzoyloxy-3-[(phenylethyloxy)methyl]-1H-indole-2-carboxylate (**1ez**)

Use of SnCl_2_·2H_2_O (82.8 mg, 0.37 mmol, 3.3 eq), 2-phenylethyl alcohol (27 μL, 0.22 mmol, 2.0 eq), and **2** (30 mg, 0.11 mmol, 1.0 eq) for 1.5 h at 40 °C, then use of DBU (233 μL, 1.56 mmol, 14.0 eq) and benzoyl chloride (26 μL, 0.22 mmol, 2.0 eq) for 2 h in general procedure afforded the title compound **1ez** (16.3 mg, 32%) as a pale-yellow solid. Mp 112–113 °C; *R*_f_ 0.44 (1:4 EtOAc/hexanes); HPLC t*_R_* 32.0 min; UV–Vis (CH_3_CN-H_2_O) λ_max_ 235, 296 nm; ^1^H NMR (300 MHz, CDCl_3_) *δ* 8.22 (d, *J* = 7.2 Hz, 2H, Ar), 7.73 (t, *J* = 7.4 Hz, 1H, Ar), 7.58 (t, *J* = 7.7 Hz, 2H, Ar), 7.31–7.18 (m, 8H, Ar), 5.24 (s, 2H, C(3)CH_2_O), 3.84 (t, *J* = 7.4Hz, 2H, OC*H*_2_CH_2_), 3.79 (s, 3H, CO_2_CH_3_), 2.97 (t, *J* = 7.4 Hz, 2H, OCH_2_C*H*_2_); ^13^C NMR (75 MHz, CDCl_3_) *δ* 164.6 (NOC(O)), 160.3 (*C*O_2_CH_3_), 139.3, 137.5, 134.9, 130.6, 129.2, 129.1, 129.0, 128.4, 127.1, 126.4, 126.2, 125.5, 123.7, 120.4, 119.0, 108.3 (Ar), 71.4 (O*C*H_2_CH_2_), 62.0 (C(3)*C*H_2_O), 52.3 (CO_2_*C*H_3_), 36.5 (OCH_2_*C*H_2_); MS m/z 463 [M]^+^; HRMS (+ESI) calcd for C_26_H_22_ClNO_5_ [M]^+^ 463.1187, found 463.1188.

Methyl 4-chloro-1-acetoxy-3-[(cyclohexyloxy)methyl]-1H-indole-2-carboxylate (**1fx**)

Use of SnCl_2_·2H_2_O (82.8 mg, 0.37 mmol, 3.3 eq), cyclohexanol (23 μL, 0.22 mmol, 2.0 eq), and **2** (30 mg, 0.11 mmol, 1.0 eq) for 2 h at 40 °C, then use of DBU (233 μL, 1.56 mmol, 14.0 eq) and acetic anhydride (32 μL, 0.22 mmol, 2.0 eq) for 2 h in general procedure afforded the title compound **1fx** (10.2 mg, 24%) as a pale-yellow solid. Mp 80–82 °C; *R*_f_ 0.42 (1:2 EtOAc/hexanes); HPLC t*_R_* 24.9 min; UV–Vis (CH_3_CN-H_2_O) λ_max_ 235, 296 nm; ^1^H NMR (300 MHz, CDCl_3_) *δ* 7.29–7.13 (m, 3H, Ar), 5.14 (s, 2H, C(3)CH_2_O), 3.94 (s, 3H, CO_2_CH_3_), 3.51–3.44 (m, 1H, OCH), 2.43 (s, 3H, OC(O)CH_3_), 2.06–1.25 (m, 10H, (CH_2_)_5_); ^13^C NMR (75 MHz, CDCl_3_) *δ* 168.6 (NOC(O)), 160.4 (*C*O_2_CH_3_), 137.1, 129.0, 127.0, 125.0, 123.6, 120.2, 119.0, 108.0 (Ar), 77.9 (OCH), 59.4 (C(3)*C*H_2_O), 52.3 (CO_2_*C*H_3_), 32.6, 26.1, 24.6 (OCH*C*H_2_*C*H_2_*C*H_2_), 18.2 (N(1)OC(O)*C*H_3_); MS m/z 379 [M]^+^; HRMS (+ESI) calcd for C_19_H_22_ClNO_5_ [M]^+^ 379.1187, found 379.1183.

Methyl 4-chloro-1-pivaloyloxy-3-[(cyclohexyloxy)methyl]-1H-indole-2-carboxylate (**1fy**)

Use of SnCl_2_·2H_2_O (82.8 mg, 0.37 mmol, 3.3 eq), cyclohexanol (23 μL, 0.22 mmol, 2.0 eq), and **2** (30 mg, 0.11 mmol, 1.0 eq) for 2 h at 40 °C, then use of DBU (233 μL, 1.56 mmol, 14.0 eq) and pivaloyl chloride (28 μL, 0.22 mmol, 2.0 eq) for 1.5 h in general procedure afforded the title compound **1fy** (13.9 mg, 30%) as a pale-yellow solid. Mp 98–100 °C; *R*_f_ 0.63 (1:2 EtOAc/hexanes); HPLC t*_R_* 34.7 min; UV–Vis (CH_3_CN-H_2_O) λ_max_ 212, 234, 296 nm; ^1^H NMR (300 MHz, CDCl_3_) *δ* 7.24–7.15 (m, 2H, Ar), 7.01 (d, *J* = 7.9 Hz, 1H, Ar), 5.13 (s, 2H, C(3)CH_2_O), 3.89 (s, 3H, CO_2_CH_3_), 3.48–3.41 (m, 1H, OCH), 1.44 (s, 9H, (CH_3_)_3_), 2.10–1.23 (m, 10H, (CH_2_)_5_); ^13^C NMR (75 MHz, CDCl_3_) *δ* 175.7 (NOC(O)), 160.3 (*C*O_2_CH_3_), 137.2, 129.0, 126.9, 125.7, 123.5, 120.5, 119.3, 107.8 (Ar), 77.7 (OCH), 59.4 (C(3)*C*H_2_O), 52.2 (CO_2_*C*H_3_), 38.8 (O*C*(CH_3_)_3_), 32.6, 26.1, 24.6 (OCH*C*H_2_*C*H_2_*C*H_2_), 27.4 (OC(*C*H_3_)_3_); MS m/z 421 [M]^+^; HRMS (+ESI) calcd for C_22_H_28_ClNO_5_ [M]^+^ 421.1656, found 421.1655.

Methyl 4-chloro-1-benzoyloxy-3-[(cyclohexyloxy)methyl]-1H-indole-2-carboxylate (**1fz**)

Use of SnCl_2_·2H_2_O (82.8 mg, 0.37 mmol, 3.3 eq), cyclohexanol (23 μL, 0.22 mmol, 2.0 eq), and **2** (30 mg, 0.11 mmol, 1.0 eq) for 2 h at 40 °C, then use of DBU (233 μL, 1.56 mmol, 14.0 eq) and benzoyl chloride (26 μL, 0.22 mmol, 2.0 eq) for 2 h in general procedure afforded the title compound **1fz** (13.1 mg, 27%) as a pale-yellow solid. Mp 96–97 °C; *R*_f_ 0.52 (1:2 EtOAc/hexanes); HPLC t*_R_* 33.8 min; UV–Vis (CH_3_CN-H_2_O) λ_max_ 235, 296 nm; ^1^H NMR (300 MHz, CDCl_3_) *δ* 8.22 (d, *J* = 7.2 Hz, 2H, Ar), 7.73 (t, *J* = 7.4 Hz, 1H, Ar), 7.57 (t, *J* = 7.9 Hz, 2H, Ar), 7.29–7.17 (m, 8H, Ar), 5.20 (s, 2H, C(3)CH_2_O), 3.83 (s, 3H, CO_2_CH_3_), 3.58–3.47 (m, 1H, OCH), 2.08–1.22 (m, 10H, (CH_2_)_5_); ^13^C NMR (75 MHz, CDCl_3_) *δ* 164.7 (NOC(O)), 160.4 (*C*O_2_CH_3_), 137.6, 134.9, 130.6, 129.2, 129.0, 127.1, 126.4, 125.5, 123.7, 120.5, 119.7, 108.3 (Ar), 77.9 (OCH), 59.4 (C(3)*C*H_2_O), 52.3 (CO_2_*C*H_3_), 32.6, 26.1, 24.6 (OCH*C*H_2_*C*H_2_*C*H_2_); MS m/z 441 [M]^+^; HRMS (+ESI) calcd for C_24_H_24_ClNO_5_ [M]^+^ 441.1343, found 441.1340.

## 4. Conclusions

We reported the studies on one-pot synthesis of new 1-acyloxyindoles **1** through four-step reactions. With substrate **2** obtained by a two-step synthetic sequence, we performed the reactions using SnCl_2_·2H_2_O as a reducing agent and alcohol (R^1^OH) as a nucleophile through reduction, intramolecular addition, and nucleophilic 1,5-addition, affording intermediate 1-hydroxyindole **8**. Subsequent acylation of **8** using acetic anhydride or acyl chlorides (R^2^COX) in a basic condition provided target compound 1-acyloxyindoles **1**. Optimization of the reaction conditions was established as follows: 1) conjugate ketoester **2** (1.0 eq), SnCl_2_·2H_2_O (3.3 eq), and ROH (2.0 eq) in DME at 40 °C; and 2) DBU (14.0 eq) and acetic anhydride or acyl chloride (2.0 eq) at room temperature. Consequently, using the optimized conditions, 21 examples of new 1-acyloxyindole derivatives were successfully synthesized in modest yields (Y = 24–35%) through one-pot reaction of a four-step sequence.

## Data Availability

The data presented in this study are available in insert article or Appendix A here. Samples of compounds **1ax**–**1fz** are available from the authors.

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
