# Peer review of "Syntheses of New Multisubstituted 1-Acyloxyindole Compounds"

_molecules, 2022, doi:10.3390/molecules27196769_

Round 1
Reviewer 1 Report
The manuscript is generally well-written and structured. Although the synthesis of 1-hydroxyindoles 8 has already been published in reference 30, the manuscript contains quite interesting results of sufficient novelty, which deserve to be published after some minor corrections and modifications. The introduction also gives a nice overview on the different biological activities of this class of compounds.
Although the yield of the 21 newly prepared 1-acyloxyindoles is in general only mediocre, the lability of these compounds justifies this finding. By the way, a statement whether 1-acyloxyindoles are only labile in solution or also in the solid purified state, would be appropriate.
All derivatives are sufficiently characterized. Although the English usage is very satisfying, some sentences need a minor improvement. This can however be done by the language-editing service of MDPI.
After taking into consideration of these few remarks, the manuscript is suitable for acceptance:
1) The abstract should highlight the purpose of the of the article and describe briefly the main methods used for the synthesis of newly compounds. The optimized reaction conditions details for the prepared heterocycles should be removed.
2) In line 74, the authors indicate that "The result of synthesis of 2 was slightly improved compared to the previous results [30]". The authors should provide more information about the synthetic procedure of compound 2 in the experimental section.
3) Although the manuscript is well-documented with the relevant literature., I suggest adding these references:
i) Studies on tertiary amine oxides. LXXIX. Reactions of presence of acylating agents 2-ethoxycarbonyl-11-hydroxyindole in the presence of acylating agents
Nagayoshi, Tsuyoshi; Saeki, Seitaro; Hamana, Masatomo Chemical & Pharmaceutical Bulletin (1984), 32(9), 3678-82 |
Abstract: The Vilsmeier-Haack reaction of indole derivative I did not give 3-formylindoles but gave 3-chloro-2 ethoxycarbonylindole (II ; R = Cl) (IIII ). The electrophilic reaction of I with quinoline 1-oxide in the presence of Bz Cl or TsCl also failed, I benzoate or III being formed resp. The reaction of I with TsCl, BzCl, or Ac2O gave the corresponding acyloxyindoles (III ; R = TsO, BzO, AcO) in low yields. Treatment of I with TsCl and 1-morpholinocyclohexane in pyridine-DMF afforded IV after hydrolysis of the reaction mixture.
ii) This reference may be complementary to ref 21: The C-33 acylation of-hydroxyindoles
Chirkova, Zhanna V.; Kabanova, Mariya V.; Filimonov, Sergey I.; Abramov, Igor G.; Samet, Alexander V.; Stashina, Galina A.; Suponitsky, Kyrill Yu. Tetrahedron Letters (2017), 58(8), 755-757 |
The C-33 acylation of 1-hydroxyindole derivatives with aliphatic acid anhydrides was accomp lished in high yields using BF3 ·Et2O as a promoter, affording the corresponding 3-acyl-1-acyloxyindoles which in turn were readily hydrolyzed to the corresponding 3 acyl-1-hydroxyindole derivatives
iii) Preparation of 6-benzyloxyindoles
Nobayashi, Eiji; Kamata, Kimiko; Tano, Shuzo Japan, JP2004284978 A 2004-10-14 |
Title compounds I [R 1 = H, C 1-12 (halo)alkyl, C 2-12 alkenyl, C 2-12 alkynyl, C 1-12 alkoxy, benzyl, Ph; X = benzyl] are prepared by hydrolysis of I [X = COR 2 ; R 2 = C 1-12 (halo)alkyl, C 2-12 alkenyl, C 2-12 alkynyl, C 1-12 alkoxy, benzyl, Ph] in the presence of bases or acids and etherification of I (X = H) with benzyl halides. I (R 1 = Me, X = Ac) was treated with Na2CO3 in aqueous MeOH at 20° for 2 h and 1 etherified with PhCH2Cl in Me2CO in the presence of Na 2 CO 3 at 50° for 20 h to give 81% I (R = Me, X = benzyl), hydrolysis of which gave 89% 6-benzyloxyindole.
4) The authors should discuss at least the spectroscopic features of one selected derivative within the manuscript and indicate the consistence of the compounds (solids, oils , coloration) and not hide this information in the experimental section.
5) Personally, I think that the use of colored rection schemes would sometimes helpful to follow them at a first glance.
Reviewer 2 Report
1. The reaction only done in small scale (0.06-0.18 mmol scale). To show the generality of this methodology, better to show scalability of this reaction (at least perform, one reaction in 1 mmol scale).
2. Some of the compounds melting points should be reported as a range, since that observable is indicative of the purity of the sample being measured.
3. In HRMS data, Recheck the compounds calculated [M]+ mass, when compared with corresponding observed (found) mass.
